# NLRX1 Prevents M2 Macrophage Polarization and Excessive Renal Fibrosis in Chronic Obstructive Nephropathy

**DOI:** 10.3390/cells13010023

**Published:** 2023-12-21

**Authors:** Ye Liu, Lotte Kors, Loes M. Butter, Geurt Stokman, Nike Claessen, Coert J. Zuurbier, Stephen E. Girardin, Jaklien C. Leemans, Sandrine Florquin, Alessandra Tammaro

**Affiliations:** 1Department of Critical Care Medicine, Zhongnan Hospital of Wuhan University, Wuhan 430071, China; 2Department of Pathology, Amsterdam Infection & Immunity, Amsterdam UMC, University of Amsterdam, 1105 AZ Amsterdam, The Netherlands; 3Department of Anesthesiology, Amsterdam UMC, University of Amsterdam, 1105 AZ Amsterdam, The Netherlands; 4Department of Laboratory Medicine and Pathobiology, University of Toronto, Toronto, ON M5S 1A1, Canada

**Keywords:** NLRX1, macrophage polarization, OXPHOS, renal fibrosis

## Abstract

Background: Chronic kidney disease often leads to kidney dysfunction due to renal fibrosis, regardless of the initial cause of kidney damage. Macrophages are crucial players in the progression of renal fibrosis as they stimulate inflammation, activate fibroblasts, and contribute to extracellular matrix deposition, influenced by their metabolic state. Nucleotide-binding domain and LRR-containing protein X (NLRX1) is an innate immune receptor independent of inflammasomes and is found in mitochondria, and it plays a role in immune responses and cell metabolism. The specific impact of NLRX1 on macrophages and its involvement in renal fibrosis is not fully understood. Methods: To explore the specific role of NLRX1 in macrophages, bone-marrow-derived macrophages (BMDMs) extracted from wild-type (WT) and NLRX1 knockout (KO) mice were stimulated with pro-inflammatory and pro-fibrotic factors to induce M1 and M2 polarization in vitro. The expression levels of macrophage polarization markers (*Nos2*, *Mgl1*, *Arg1*, and *Mrc1*), as well as the secretion of transforming growth factor β (TGFβ), were measured using RT-PCR and ELISA. Seahorse-based bioenergetics analysis was used to assess mitochondrial respiration in naïve and polarized BMDMs obtained from WT and NLRX1 KO mice. In vivo, WT and NLRX1 KO mice were subjected to unilateral ureter obstruction (UUO) surgery to induce renal fibrosis. Kidney injury, macrophage phenotypic profile, and fibrosis markers were assessed using RT-PCR. Histological staining (PASD and Sirius red) was used to quantify kidney injury and fibrosis. Results: Compared to the WT group, an increased gene expression of M2 markers—including *Mgl1* and *Mrc1*—and enhanced TGFβ secretion were found in naïve BMDMs extracted from NLRX1 KO mice, indicating functional polarization towards the pro-fibrotic M2 subtype. NLRX1 KO naïve macrophages also showed a significantly enhanced oxygen consumption rate compared to WT cells and increased basal respiration and maximal respiration capacities that equal the level of M2-polarized macrophages. In vivo, we found that NLRX1 KO mice presented enhanced M2 polarization markers together with enhanced tubular injury and fibrosis demonstrated by augmented TGFβ levels, fibronectin, and collagen accumulation. Conclusions: Our findings highlight the unique role of NLRX1 in regulating the metabolism and function of macrophages, ultimately protecting against excessive renal injury and fibrosis in UUO.

## 1. Introduction

Tissue fibrosis, defined by the excessive accumulation of extracellular matrix (ECM), is the final common pathway of chronic diseases, including chronic kidney diseases (CKD), and is a leading cause of morbidity and mortality worldwide [1].

Fibrotic kidney disease is characterized by the disruption of tubular integrity, dedifferentiation of tubules, loss of peritubular capillaries, interstitial inflammation, and accumulation of (myo)fibroblast, ultimately leading to fibrogenesis [2,3]. Renal fibrosis, regardless of the primary renal condition, is the shared pathogenic pathway culminating in CKD. Therefore, it is imperative to uncover factors that drive the onset and progression of renal fibrosis.

Recent evidence highlights the critical role of innate immunity in renal fibrosis [4]. Macrophages, the prevalent immune cell type in damaged kidney tissue, actively contribute to renal fibrosis during CKD [5,6,7]. Depending on their local environment, macrophages can assume different activation states, polarizing towards either pro-inflammatory (M1) or alternatively activated (M2) forms, each with distinct functions [8]. M1 macrophages drive inflammation during kidney injury, while M2 macrophages promote tissue repair and, in specific cases, contribute to fibrosis persistence [9,10,11]. Our research and others’ findings have previously established the essential involvement of innate immune sensors, like triggering receptors expressed on myeloid cells (TREM1), toll-like receptor 4 (TLR4), and stimulator of interferon genes (STING), in inducing renal tubulointerstitial fibrosis post-UUO [12,13,14,15]. Notably, we have identified S100A8/A9 (TLR4 ligand) as a key regulator controlling macrophage-polarization-mediated renal fibrosis after acute kidney injury, bridging the gap between innate immunity and macrophage activation states in renal fibrogenesis [16]. Studies have demonstrated that the downregulation of TLR4, facilitating the shift from the M1 to the M2 phenotype, can decrease fibrogenesis [17]. Likewise, activation of the NLR family pyrin domain containing 3 (NLRP3) and toll-like receptor 9 (TLR9) regulate macrophage polarization and renal fibrosis in different preclinical models [18,19,20].

Macrophage polarization is dependent on metabolic regulation. Extensive research has established that M1 macrophages have an enhanced glycolytic metabolism, while M2 macrophages manifest high metabolic dependency on mitochondrial oxidative phosphorylation (OXPHOS) to fulfill their energy demands [21]. Recent studies have revealed that the innate immune receptor STING is engaged in the repolarization of M2 towards M1 by reducing OXPHOS and promoting glycolysis. This highlights the pivotal role of innate immune sensors in shaping macrophage polarization through metabolic regulation [22,23,24]. 

In our previous work, we demonstrated the significant influence of nucleotide-binding domain and LRR-containing protein X (NLRX1), an innate immune receptor uniquely localized to the mitochondrial matrix, in modulating mitochondrial activities in hepatocytes and tubular epithelial cells (TECs). NLRX1 mitigates OXPHOS during cell stress and tissue injury [25,26]. In addition, in our earlier work, we noted an increase in macrophage influx in NLRX1 knockout (KO) kidneys after acute ischemic injury, suggesting a potential role of NLRX1 in macrophages during renal repair [26]. Furthermore, studies have linked the downregulation of NLRX1 to the severity of chronic obstructive pulmonary disease, suggesting its possible involvement in fibrosis pathogenesis chronic diseases [27]. 

As such, we hypothesize that NLRX1 may play a crucial role in the metabolic regulation of macrophages, influencing their activation state and function. This, in turn, could significantly impact the progression of renal fibrosis and chronic renal disease. Through a comprehensive approach involving in vitro and in vivo studies we demonstrate that NLRX1-deficient macrophages exhibit an intrinsic enhanced polarization toward the pro-fibrotic M2 subtype, which heavily relies on OXPHOS metabolism, unlike wild-type (WT) macrophages. Moreover, the absence of NLRX1 significantly enhances levels of the pro-fibrotic cytokine transforming growth factor β (TGFβ) in bone-marrow-derived macrophages (BMDM). To validate these in vitro findings, we employed an in vivo model of progressive renal fibrosis. Both WT and NLRX1 KO mice were subjected to unilateral ureteral obstruction (UUO) [25]. In line with our in vitro findings, we observed that the lack of NLRX1 enhances TGFβ levels, exacerbating renal injury and interstitial fibrosis compared to WT mice. Our results underscore the significant role of NLRX1 in the immune–metabolic regulation of macrophages, controlling both their polarization and function and as a protective factor against excessive renal fibrosis. 

## 2. Materials and Methods

### 2.1. Animal Experimental Procedures

NLRX1 KO mice were generated as described previously [28] and bred at our institute. Age- and gender-matched C57BL6/J WT male mice were obtained from Charles River (Maastricht, The Netherlands). For bone marrow isolation, the femur and tibia from 6–10 week old WT and NLRX1-KO mice were collected after sacrificing the mice under 4% isoflurane/O_2_ followed by cervical dislocation. Subsequently, the bones were further processed as described later. WT and NLRX1-KO female mice were subjected to UUO surgery as described below. Briefly, prior to surgery, all mice received analgesics (subcutaneous injection of 50 µg/kg buprenorphine (Temgesic, Schering-Plough, Amstelveen, The Netherlands), and subsequently, the right ureter was exposed following an abdominal incision and ligated using non-dissolving sutures under 2% isoflurane/O_2_ anesthesia. The abdomen was closed in two layers. Post-surgery mice were sacrificed via heart puncture under 4% isoflurane/O_2_ followed by cervical dislocation. The contralateral non-obstructed kidney of day 1 subjected mice served as the control. Kidneys were collected, dissected, and snap-frozen in liquid nitrogen and subsequently stored at −80 °C or fixed in formalin until further processing. All animal procedures were ethically approved under DPA 4AD-1 by the Animal Care and Use Committee of the AMC Amsterdam and were conducted in compliance with the ARRIVE guidelines (NC3Rs). 

### 2.2. Histology and Immuno-Histochemistry

For paraffin embedding, 24 h-formalin-fixed kidneys were processed in a Tissue-Tek VIP (Sakura, Alphen a/d Rijn, The Netherlands). For histopathology scoring, tissue sections (4 µm) were de-waxed and PAS-D stained. For total collagen staining, de-waxed tissue sections (4 µm) were incubated with 0.2% Picro-Sirius-Red solution (pH 2.0) for 1 h, followed by incubation in 0.01 M HCL. Ki67 staining was performed on de-waxed tissue sections (4 µm) that were treated with 0.3% hydrogen peroxide in methanol for 15 min. Sections were boiled for 10 min in citrate buffer (pH 6.0) and overnight at 4 °C incubated with primary antibody Rabbit-anti-Mouse-Ki67 (cat no. RM-9106-s0, ThermoFisherScientific, Waltham, MA, USA) in antibody diluent (ThermoFisherScientific). Sections were incubated with peroxidase-conjugated secondary antibodies (AgilentDako, Santa Clara, CA, USA) for 30 min. Subsequently, sections were stained with 3,3-diaminobenzidine (DAB), and nuclei were counterstained with hematoxylin.

### 2.3. Tubular Injury Scoring and Immuno-Histochemistry Analysis

Tubular injury was defined as epithelial flattening, tubular dilatation, and brush border loss. The percentage of tubular injury was scored by the nephropathologist (SF) in a blinded fashion in ten randomly chosen, non-overlapping fields (200× magnification) using a 5-point tubular injury scoring system: Score 0: Normal, no tubular injury; Score 1: Mild, involvement of less than 25% of the cortex; Score 2: Moderate, involvement of 25–50% of the cortex; Score 3: Severe, involvement of 50–75% of the cortex; Score 4: Extensive damage, involvement of over 75% of the cortex [29]. The number of proliferating tubular cells (Ki67) in the renal cortical region was determined in 10 randomly chosen, non-overlapping fields (200× magnification). Picro-Sirius-Red positive staining was quantified digitally in 10 randomly chosen, non-overlapping fields in the renal cortical region (200× magnification) using ImageJ software version 1.5.

### 2.4. Macrophage Culture and Differentiation

Isolation and culturing of BMDMs were performed as partly described before [30]. Briefly, on day 0, mononuclear phagocyte progenitor cells were collected by flushing the bone marrow from the femur and tibia with a syringe containing ice-cold sterile PBS and grown in bone marrow macrophage culture medium (BMCM) that contains: RPMI 1640 medium supplemented with 10% FCS, 2 mM L-glutamine and 100 U/mL Penicillin/Streptomycin (all from ThermoFisherScientific), and 15% LCM (L929 conditioned medium) and cultured for 8 days at 37 °C and 5% CO_2_ in two 145 × 20-mm uncoated petri dishes (Greiner, Kremsmünster, Austria). On day 3, fresh BMCM was added, and on day 6 the medium was replaced. On day 8, BMDM were detached using sterile Lidocaine (Sigma-Aldrich, St. Louis, MO, USA) solution (4 mg/mL sterile PBS), harvested, and counted using a hemocytometer. A total of 5 × 10^4^ cells per well in a XF96-well assay plates and 1 × 10^5^ cells per well in a 24-well plate were used. BMDMs were incubated in BMCM supplemented with LPS (10 ng/mL; Sigma-Aldrich) and IFNγ (50 ng/mL; ProSpec, Mount Pleasant, SC, USA) for 24 h to initiate M1 activation, or IL-4 (50 ng/mL; ProSpec) for 24 h to initiate M2 activation. For the tubule-damaged microenvironment exposure experiments, BMDMs were incubated in medium containing 0.15 mg/mL of supernatant from damaged proximal tubular epithelial cells.

### 2.5. In Vitro Assays with Damaged Proximal Tubular Epithelial Cells

Conditionally immortalized murine proximal tubular epithelial cells (IM-pTECs) were generated as previously described [31]. IM-pTECs were cultured in DMEM/F12 medium with 10% FCS, 5 ug/mL insulin and transferrin, 5 ng/mL sodium selenite (all from ThermoFisherScientific), 20 mg/mL triiodo-thyrionine (Sigma-Aldrich), 50 ng/mL hydrocortisone (Sigma-Aldrich), and 5 ng/mL prostaglandin E1 (Sigma-Aldrich) with 2 mM L-glutamine and 100 U/mL Penicillin/Streptomycin (both from ThermoFisherScientific). IM-PTECs were maintained at 33 °C in the presence of 10 ng/mL IFNγ (ProSpec) and differentiated at 37 °C without IFNγ for one week, resulting in loss of SV40 expression [31]. After one week of differentiation, cells were used for experimentation. Supernatant from damaged IM-pTECs was obtained via repeated freezing and thawing cycles, as described earlier by Sauter [32]. Total protein concentration in the supernatant was determined by BCA (Sigma-Aldrich). The supernatant was filtered through 0.22 µm filter and used to stimulate M0 bone marrow-derived macrophages at 0.15 mg/mL.

### 2.6. Enzyme-Linked Immunoabsorbance Assay (ELISA)

Frozen kidney tissues 5% (*w*/*v*) were homogenized in Greenberger lysis buffer (GLB) (300 mM NaCl, 30 mM Tris, 2 mM MgCl_2_, 2 mM CaCl_2_, 1% (*v*/*v*) Triton X-100, pH set at 7.4, supplemented with Protease Inhibitor Cocktail II (Sigma-Aldrich). In homogenates and BMDM supernatants TNFα, IL-6 and active TGFβ levels were determined via duo set ELISAs (R&D Systems, Abingdon, UK) performed according to the supplied protocol. ELISA data measured in BMDM supernatants were adjusted for cell input using the CyQUANT^®^ Cell Proliferation Assay Kit (ThermoFisherScientific) according to the supplier’s protocol. ELISA data measured in kidney homogenates were adjusted for total protein concentration as determined by BCA (Sigma-Aldrich). 

### 2.7. RNA Isolation and Real Time PCR

Total RNA was extracted from BMDM and frozen kidney tissues using TriReagent (Sigma-Aldrich) followed by chloroform extraction and isopropanol precipitation and converted to cDNA. cDNA was synthesized using the M-MLV reverse transcriptase (ThermoFisherScientific). Transcription was analyzed via real-time quantitative PCR (qPCR) on a Roche LightCycler 480 using sensiFAST SYBR master mix (Bioline reagents, London, UK). Relative mitochondrial DNA abundance by 16 SrRNA/Hexokinase 2 real time quantitative PCR ratio was quantified as described before [33]. In short, DNA was extracted using the standard TriReagent DNA extraction according to the manufacturer’s protocol, and the resulting genomic DNA was used for qPCR. qPCR primers were synthesized using Eurogentec (Maastricht, The Netherlands) and described in the Appendix A.

### 2.8. Cell Assays, Biochemical Analysis, and Enzyme Activity

Glucose and lactate levels in the medium were determined through routine diagnostic procedures by the department of Clinical Chemistry of the AMC Amsterdam. For the CS activity assay BMDM cells were collected and lysed in 0.5% (*w*/*v*) Triton X-100, and activity was determined as described previously [34]. For ATP measurements, BMDM were cultured in 24-well plates. One hour prior, the assay control cells were incubated with 5 mM 2-Deoxy-D-glucose and 2.5 µm AntimycinA (Sigma-Aldrich). ATP levels were determined using a kit (ThermoFisherScientific) according to the manufacturer’s protocol. Cs activity, glucose, lactate, and ATP values were adjusted for total protein concentration as determined using BCA (Sigma-Aldrich). Cellular OCR and ECAR were determined via real-time extracellular flux analysis using the XFe96 Seahorse Extracellular Flux Analyzer (Agilent, Santa Clara, CA, USA) in BMDM plated and polarized in XF 96-well assay plates 48 h before the assay. OCR and ECAR were determined with the extracellular flux assay according the protocol of Van den Bossche et al. [30]. ECAR and OCR values were adjusted for cell input using the CyQUANT^®^ Cell Proliferation Assay Kit (ThermoFisherScientific) according to the supplier’s protocol.

### 2.9. Statistical Analysis 

Data are presented as mean ± standard error of the mean (SEM). The non-parametric Mann Whitney U test was performed for two-group comparison. For all analyses, values of *p* ≤ 0.05 were considered significant. All statistical analyses were performed using GraphPad Prism9 (GraphPad Software, San Diego, CA, USA).

## 3. Results

### 3.1. NLRX1 Deficiency Leads to Polarization towards the Pro-Fibrotic M2 Subtype in BMDM

In previous works, it has been reported that energy metabolism is under the control of NLRX1 in epithelial cells, including hepatocytes and renal tubular cells, during cell stress and tissue injury [25,26]. To further explore its effect on macrophages, which are present in the kidney as resident cells [35] and play an essential role in chronic kidney injury, we specifically investigate the contribution of NLRX1 in macrophage polarization of BMDMs obtained from WT and NLRX1 KO mice. Macrophages can be categorized in vitro into naïve (M0), classically activated (M1; using LPS + IFNγ), or alternatively activated (M2; using IL4) macrophages. Compared to M0 macrophages, we found that *Nlrx1* mRNA expression levels significantly decreased in macrophages polarized to M1 and showed a trend to be lower when polarized to M2 (Figure 1A). Next, we determined the expression of several genes involved in pro-inflammatory M1 (*Nos-2*) and pro-fibrotic M2 (*Arg-1*, *Mgl-1*, and *Mrc-1*) macrophage polarization/activation [9,36]. We confirmed that *Nos-2* was expressed by M1 polarized cells, although no large difference was found between WT and NLRX1 KO groups (Figure 1B). *Mgl-1, Mrc-1*, and *Arg-1* were expressed by M2 polarized cells independent from NLRX1 presence (Figure 1C–E). Interestingly, *Mgl-1* expression in M0 macrophages and *Mrc-1* in M2 were significantly increased in the absence of NLRX1 compared to WT, indicating that loss of NLRX1 does alter macrophage marker expression towards the pro-fibrotic M2 subtype (Figure 1C,D). 

Subsequently, we investigated the functional impact of NLRX1 deficiency on cytokine secretion in the different BMDM subtypes. This revealed that IL-6 and TNFα levels, pro-inflammatory cytokines that are typically secreted by M1 polarized macrophages, were similar between WT and NLRX1 KO (Figure 1F,G). Specifically, we observed that NLRX1 KO macrophages produce much more pro-fibrotic TGFβ compared to WT (Figure 1H). Active TGFβ levels secreted by M2 polarized cells were to some extent but non-significantly higher in the NLRX1 KO and WT (Figure 1H). To investigate NLRX1’s potential role in promoting M2 polarization, we subjected both WT and NLRX1 KO M0 macrophages to a pro-inflammatory environment generated by injured kidney epithelial cells. This setup mirrors how macrophages typically respond within the damaged renal microenvironment and allows us to assess whether NLRX1 influences the shift towards M2 polarization. We observed significantly increased levels of *Tgfβ* and *Mgl-1* in NLRX1 KO M0 compared to WT (Figure 1I,J), indicating a pro-fibrotic M2 characteristic of NLRX1 KO macrophages in response to the damaged TEC microenvironment. Together, these data imply that NLRX1 loss controls intrinsic macrophage polarization, leading to a functional pro-fibrotic M2 macrophage subtype.

### 3.2. NLRX1 Deficiency Promotes Mitochondrial Oxidative Metabolism in BMDM

Polarized macrophages display a distinct regulation of their bioenergetics: In-vitro-polarized M1 macrophages are metabolically supported by glycolysis, and M2 macrophages use predominantly fatty acid oxidation (FAO) and mitochondrial OXPHOS to fulfill their energy demand [37,38]. Therefore, we examined the impact of NLRX1 on macrophage metabolic functionality. We performed a metabolic characterization of different WT and NLRX1 KO BMDM subtypes M0, M1, and M2. Using real-time extracellular flux analysis, we determined the oxygen consumption rate (OCR) (Figure 2A,B) and the extracellular acidification rate (ECAR) (Figure 2C,D) of WT and NLRX1 deficient BMDMs. In line with the literature [30], WT M2 macrophages demonstrate a significant enhanced oxidative metabolism with an increased basal respiration and maximal respiratory capacity compared to WT M0 macrophages (Figure 2A). In contrast, M1 macrophages display lowered oxidative metabolism (Figure 2A) and an enhanced glycolytic metabolism (Figure 2C) compared to M0. Interestingly, NLRX1 KO M0 macrophages show a significantly enhanced OCR compared to WT cells with increased basal respiration and maximal respiration capacity that equals the level of M2 polarized cells (Figure 2A,B). This indicates that M0 macrophages absent of NLRX1 are metabolically polarized towards the M2 subtype. On the other hand, no differences in ECAR and glycolysis were observed between WT and NLRX1 KO M0 macrophages (Figure 2C,D). The glycolytic metabolism presented similar results in WT and NLRX1 KO M0 macrophages characterized by glucose consumption and lactate production (Figure 2E,F). Interestingly, adenosine triphosphate (ATP) levels are found to be significantly lower in NLRX1 KO M0 compared to WT (Figure 2G). This suggests that with increased oxygen consumption in NLRX1 KO M0 macrophages, ATP is much more consumed or alternatively that ATP synthesis is less efficient compared to the controls.

To further understand the role of NLRX1 in mitochondrial metabolism, we examined different transcript encoding proteins involved in mitochondrial OXPHOS (*Ndufa2*, *Ndufb3*, *Cyt C*, *Cox4i1*), FAO (*Cpt1b*, *Acadm*, *Acadl*), and mitochondrial biogenesis (*Tfam*) in WT and NLRX1 KO M0 macrophages, and no difference was observed in the gene expression level. (Figure 2H) Moreover, the mitochondrial content as reflected by mtDNA abundance (mitochondrial genome to nuclear genome ratio (Mt/N)) (Figure 2I) and citrate synthase (CS) activity (Figure 2J) also showed similar results in MT and NLRX1 KO M0 macrophages. Together, these data show the importance of NLRX1 for macrophage metabolism and NLRX1 deficiency in unpolarised M0 macrophages leads towards the pro-fibrotic M2 subtype metabolic characteristics without affecting mitochondrial content.

### 3.3. NLRX1 Deficiency Aggravates Renal Injury following UUO

Given the involvement of macrophages in the development of renal fibrosis [5,6,7], we next investigated the role of NLRX1 in the UUO model. We found that *Nlrx1* mRNA levels were significantly decreased 1 and 3 days post-UUO compared to the control group (Figure 3A). Therefore, we continued by subjecting WT and NLRX1 KO mice to UUO. Histological evaluation of kidneys revealed a progressive increase in tubular injury, including epithelial flattening, brush border loss, and tubular dilatation in both WT and NLRX1 KO mice (Figure 3B,C). Specifically, NLRX1 KO mice developed significantly more tubular injury by day 3. In addition, the mRNA expression levels of *Kim-1* as a marker for proximal tubular damage and *Ngal* for distal tubular damage were examined. In line with this, NLRX1 KO mice had enhanced *Kim-1* levels 3 days after UUO compared to WT (Figure 3D), whereas *Ngal* was increased after UUO with no differences between WT and NLRX1 KO (Figure 3E). The increase in tubular damage in the NLRX1 KO mice was associated with an upregulated proliferation detected by an increase in Ki67+ TECs (Figure 3F,G). Together, these data show that renal *Nlrx1* mRNA expression is significantly decreased during the early post-UUO time points and that NLRX1 absence aggravates UUO-induced tubular injury, especially proximal tubular damage, and promotes cellular proliferation at an early stage. 

### 3.4. Absence of NLRX1 Enhances Interstitial Fibrosis following UUO

We then reasoned that the immunometabolic characteristics of macrophages lacking NLRX1, as observed in our in vitro experiment could potentially influence the progression of renal fibrosis. UUO-induced renal fibrosis is characterized by infiltration of macrophages and deposition of fibronectin and collagen in the tubulo-interstitial area [39]. We found that 3 days after UUO, *F4/80(Emr1)* mRNA expression levels were significantly more elevated in the NLRX1 KO mice compared to WT (Figure 4A), indicating increased macrophages in the kidney during the early UUO onset in the absence of NLRX1. Next, we determined the renal expression of pro-inflammatory M1 (*Nos-2*) and pro-fibrotic M2 (*Arg-1*, *Mgl-1*, and *Mrc1*) markers. *Arg-1* and *Mrc1* expressions were both significantly elevated 3 days post-UUO in NLRX1 deficient kidneys compared to WT (Figure 4B,C), whereas *Mgl-1* and *Nos-2* showed no difference between WT and NLRX1 KO mice (Figure 4D,E), indicating that due to the absence of NLRX1, pro-fibrotic macrophages are more abundant in kidneys in the early stage of UUO. Subsequently, we examined the level of active TGFβ which is typically secreted by pro-fibrotic macrophages and plays an important role in renal fibrosis [10,40,41]. We observed that 1 and 3 days following UUO, absence of NLRX1 leads to increased renal levels of active TGFβ compared to WT (Figure 4F). In addition, renal *Fibronectin* and *collagen-1* mRNA expression levels were both higher in NLRX1 KO mice 3 days post-UUO compared to WT kidneys (Figure 4G,H). Despite reaching the maximum renal injury, collagen deposition is more pronounced at a later stage of UUO [42]. To study total collagen deposition, we therefore included kidneys at later time points of UUO. As a result, the presence of total collagen as reflected by Picro-Sirius-Red (PSR) staining was more abundant in NLRX1 KO kidneys compared to WT at day 12 (Figure 4I,J). Together, these data show that renal levels of TGFβ and multiple markers of renal fibrosis are increased in the absence of NLRX1 during UUO. 

## 4. Discussion

Metabolic reprogramming of macrophages can influence the progression and resolution of renal fibrosis [6]. Understanding regulatory mechanisms involved in macrophage polarization and function is essential for developing targeted therapies to prevent renal fibrosis. Our findings highlight the pivotal role of NLRX1 in directing macrophage polarization and OXPHOS metabolism. This regulation has functional implications for the expression and secretion of TGFβ, ultimately acting as protective mechanism against tubular injury and excessive renal fibrosis in chronic obstructive nephropathy. 

It has been well investigated that macrophages react heterogeneously to the kidney microenvironment and assume different metabolic and phenotypic changes during kidney diseases [43,44]. The polarization of macrophages to a pro-fibrotic M2 phenotype is considered to play vital role during the development of renal fibrosis by producing and secreting TGFβ and fuelling a fibrotic kidney microenvironment [10]. In our study, we applied BMDMs to explore the role of NLRX1 in macrophage polarization. Our observations revealed that in comparison to M0 macrophages, *Nlrx1* expression is notably reduced in M1 and tends to be lower in M2 macrophages. The loss of NLRX1 in M0 macrophages is linked to an immunometabolic M2 phenotype. This is characterized by increased expression of the M2 markers *Mgl-1*, as well as an enhanced reliance on OXPHOS and increased functional TGFβ secretion. These findings suggest that in an unconditional microenvironment, the absence of NLRX1 steers BMDM towards M2 characteristics. However, it is worth noting that this transition does not align completely with the conventional understanding of macrophage heterogeneity based on marker expression [45].

Furthermore, when exposed to a microenvironment of damaged TECs [32], M0 macrophages lacking NLRX1 displayed similar pro-fibrotic M2 features. The significant role of innate immune sensors and their ligands in macrophage activation and polarization has been previously described [17,19,46,47]. Our findings further reinforce this notion, particularly in the context of NLRX1.

Apart from equal levels of *Nos-2* expression between WT and NLRX1 KO, it is clear from our study that NLRX1 is not essential for the M1 pro-inflammatory responses, including TNFα and IL6 cytokine release. This is in line with observations made in LPS-stimulated whole blood and BMDM, in which TNFα and IL6 expressions remained similar in WT and NLRX1-deficient cells [48]. Our study revealed that OXPHOS metabolism is increased in the absence of NLRX1 while glycolytic metabolism remains unchanged, indicating that the metabolic profile in M0 cells is reprogrammed towards a pro-fibrotic M2 instead of M1 macrophages. Moreover, significantly increased secretion of active TGFβ in NLRX1 KO M0 macrophages by unknown stimuli further suggests that the metabolic profile contributes to the pro-fibrotic functional outcome. 

The interaction between metabolic switches and immunological activity of immune cells is studied in the emerging field of immuno-metabolism. Although innate immune receptors primarily function as activators of innate immunity, there is increasing evidence that some are important in the regulation of cellular metabolic pathways as well [37]. For example, dendritic cell activation via TLR2 and TLR4 has been shown to enhance glycolysis metabolism [49,50]. The observation that NLRX1 is an essential regulator of macrophages metabolic reprogramming is in agreement with previous findings in non-immune parenchymal cells including hepatocytes and TECs [25,26]. This indicates that NLRX1 can function as a metabolic regulator in cells of both myeloid and epithelial origins. Furthermore, the regulation of energy metabolism by NLRX1 appears to be independent of the mitochondrial content, as reflected by equal mitochondrial DNA abundance, citrate synthase activity and mitochondrial gene expressions involved in OXPHOS in WT and NLRX1 KO M0 macrophages. Moreover, we found that NLRX1 deficient macrophages have a higher oxygen consumption rate but a lower level of ATP, which is in line with previous findings and indicates that NLRX1 deficiency leads to increased ATP consumption or inefficient ATP synthesis [26]. This warrants the plausibility to further study the direct role of NLRX1 in modulating OXPHOS metabolism and mitochondrial dysfunction, given the mitochondrial localization of NLRX1 and its interaction with the matrix-facing protein of the respiratory chain complex III subunit UQCRC2 [26,51]. 

Together, our data indicate that independent of the renal microenvironment, NLRX1 absence in macrophages can induce a metabolic reprogramming towards enhanced OXPHOS resulting into a functional pro-fibrotic M2 macrophage subtype. This gives NLRX1 a unique physiological role in the field of macrophage immune metabolism. Given the pivotal role of macrophages in renal fibrosis and the protective role of NLRX1 in several preclinical model of fibrosis [25,27], we investigated whether NLRX1 influenced the progression of renal fibrosis in chronic obstructive nephropathy. Here, we observed an early reduction in renal NLRX1 expression evident at 1 and 3 days post-UUO and noted an exacerbated renal tubular injury 3 days post UUO in NLRX1 KO mice compared to WT mice. The heightened tubular damage observed in KO mice corresponded with an upregulated proliferation, suggesting the initiation of renal repair. Since tissue repair often involves the participation of macrophages [45,52], we investigated the polarization status of renal macrophages in both WT and KO mice. Our findings revealed that NLRX1 KO mice subjected to UUO exhibit enhanced macrophages displaying M2 characteristics. This included a profound increase in TGFβ secretion which subsequently led to heightened renal interstitial fibrosis as indicated by increased gene expression of *fibronectin* and *collagen-1* and collagen deposition at later time points. Studies have shown that macrophages will go through a phenotype transition in response to tissue damage, and the subsequent polarization towards the M2 macrophage contributes to the development of renal fibrosis [53,54,55]. Wu et al. demonstrated alleviated renal fibrosis in UUO by inhibiting M2 polarization and reducing the secretion of pro-fibrotic factors including Arg-1, CD206 and TGFβ [55]. These results align with ours, including increased expression of *Arg-1, Mrc-1 (CD206)* and secretion of TGFβ in NLRX1 KO mice, indicating a promising role of NLRX1 in limiting pro-fibrotic macrophage polarization during the development of renal fibrosis induced by UUO. 

Our study design has some limitations that should be considered. For example, we used BMDM, which may not accurately mimic the characteristics of resident or infiltrating macrophages, as these are heavily influenced by the kidney microenvironment in the context of renal fibrosis [45,56]. Additionally, we did not perform experiments using conditional KO for NLRX1 in macrophages. This merits further investigation to comprehensively understand how NLRX1 affects macrophage metabolism and function in the progression of renal fibrosis. 

## 5. Conclusions

We identified NLRX1 as an immuno-metabolic regulator of macrophage polarization and function by modulating OXPHOS metabolism. Absence of NLRX1 polarizes macrophages towards a pro-fibrotic phenotype and aggravates renal injury and fibrosis in mice following UUO. As such, NLRX1 may be an attractive novel therapeutic target for renal fibrosis and fibrotic diseases in general.

## Figures and Tables

**Figure 1 cells-13-00023-f001:**
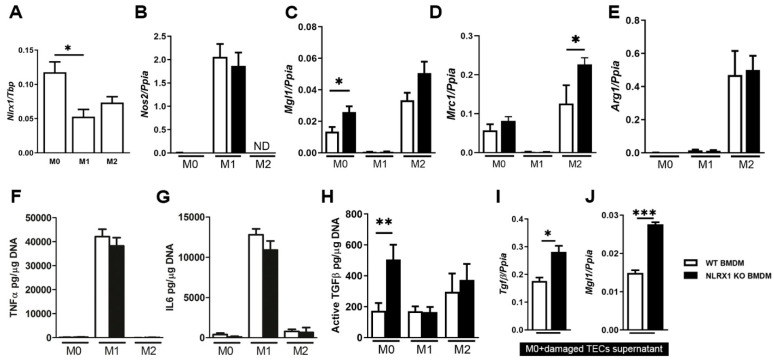
NLRX1 deficiency leads to the polarization towards the pro-fibrotic M2 subtype in bone-marrow-derived macrophages (BMDM). Parameters were determined in BMDM from wild-type (WT) and NLRX1 knockout (KO) mice. Macrophages can be categorized into naïve (M0), classically activated (M1; using LPS + IFNγ for 24 h), or alternatively activated (M2; using IL4 for 24 h) macrophages. (**A**) *Nlrx1* mRNA levels in different macrophage subsets *n* = 5–6. (**B**) *Nos2*, (**C**) *Mgl1,* (**D**) *Mrc1*, and (**E**) *Arg1* mRNA levels in different macrophage subsets *n* = 4–6 determined in duplicate. Levels of (**F**) TNFα, (**G**) IL6, and (**H**) active TGFβ in supernatants from different macrophage subsets *n* = 4 determined in triplicate. (**I**) *Tgfβ* and (**J**) *Mgl1* mRNA levels in M0 macrophages 24 h exposed to conditional supernatant from damaged tubular epithelial cells (TECs) *n* = 4 determined in duplicate. Results are representative of three independent experiments. ND, not determined. * *p* < 0.05, ** *p* < 0.01, *** *p* < 0.001.

**Figure 2 cells-13-00023-f002:**
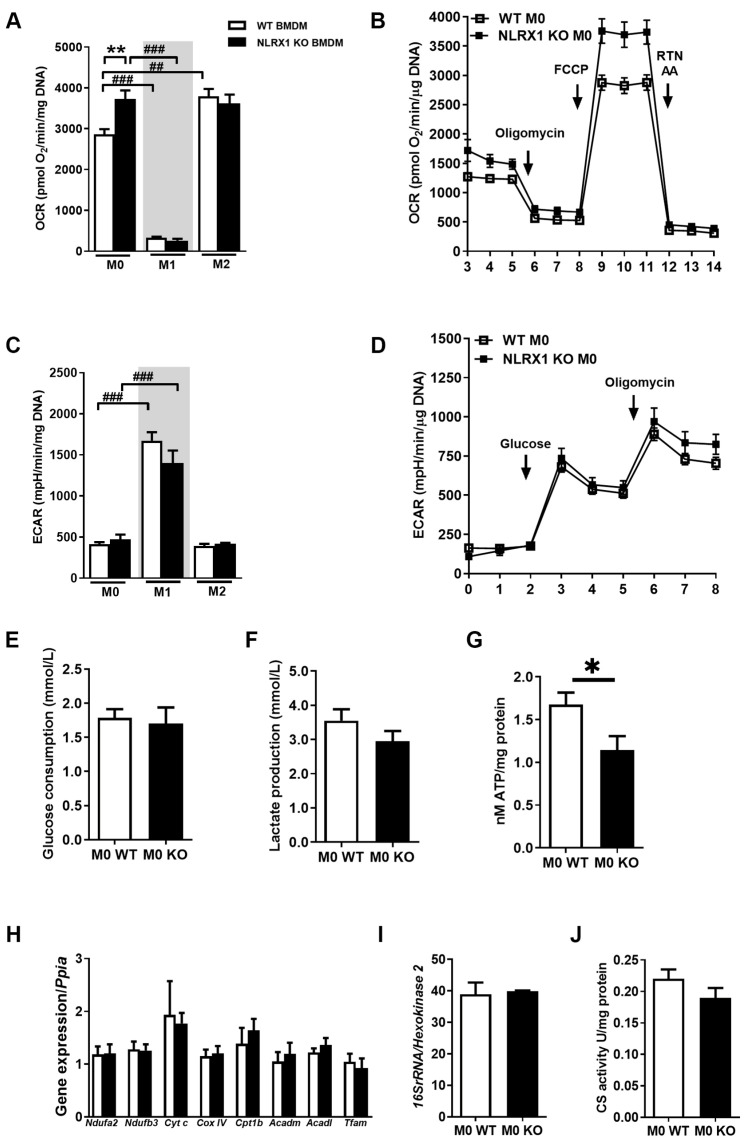
NLRX1 deficiency promotes mitochondrial oxidative metabolism in BMDM. (**A**) Maximal respiration calculated from mitochondrial oxygen consumption rate (OCR) data. (**B**) Mitochondrial OCR determined with the Seahorse XF96respirometer. X axis indicates number of measurements. Rtn, Rotenone; AA, AntimycinA. Conditions were determined six-fold. Results are representative of three independent experiments. (**C**) Glycolysis calculated from extracellular acidification rate (ECAR) data. (**D**) Glycolytic functional key parameters determined by ECAR with the Seahorse XF96respirometer. X axis indicates number of measurements. Conditions were determined six-fold. (**E**) Consumed glucose and (**F**) produced lactate levels from M0 cells after 48 h in glucose-rich medium *n* = 4–6. (**G**) Intracellular ATP levels measured in M0 macrophages adjusted to total protein contents per sample *n* = 5–6. (**H**) mRNA transcript levels of indicated genes in M0 macrophages *n* = 4. (**I**) Relative mitochondrial DNA abundance by 16S rRNA/Hexokinase2 real time quantitative PCR ratio in WT *n* = 6 and NLRX1 KO *n* = 3 M0 macrophages. (**J**) Basal mitochondrial metabolic enzymatic citrate synthase (CS) activity in M0 macrophages *n* = 6. Influence of NLRX1 deficiency versus corresponding WT cells is represented by * *p* < 0.05, ** *p* < 0.01. Influence of macrophage activation versus corresponding M0 group is displayed as ## *p* < 0.01, ### *p* < 0.001. Results are representative of three independent experiments.

**Figure 3 cells-13-00023-f003:**
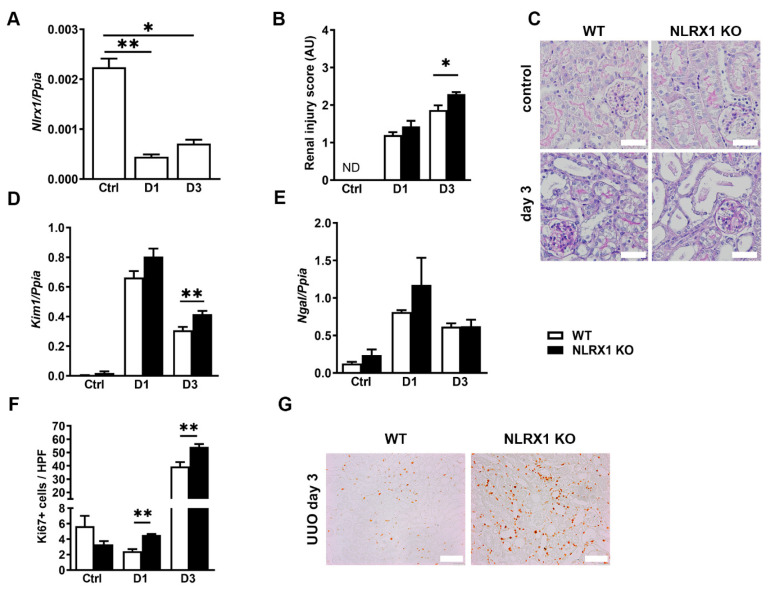
NLRX1 deficiency aggravates renal injury following unilateral ureteral obstruction (UUO). Renal injury parameters determined in wild-type (WT) and NLRX1 knockout (KO) kidneys at indicated time points in the days following UUO surgery. (**A**) *Nlrx1* mRNA levels in WT kidneys. (**B**) Semi-quantitative renal injury score. (**C**) Control- and D3-representative PASD-stained renal tissues are shown (white bar represent 50 µm). Renal (**D**) *Kim-1* and (**E**) *Ngal* mRNA levels. (**F**) Ki67 (proliferation) positive tubular cells (TECs). (**G**) Representative Ki67 stained renal tissues are shown for day 3 post-UUO (white bar represent 100 µm). * *p* < 0.05, ** *p* < 0.01.

**Figure 4 cells-13-00023-f004:**
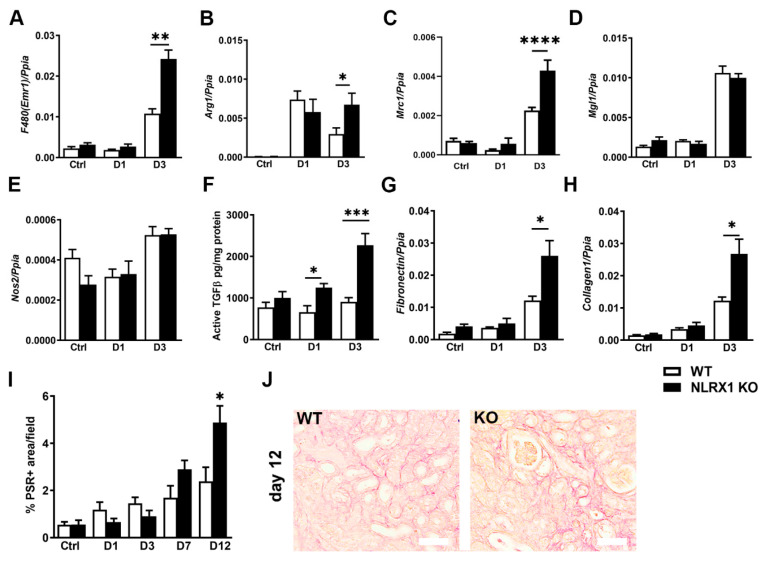
Absence of NLRX1 enhances interstitial fibrosis following UUO. (**A**) *F480(Emr1),* (**B**) *Arg1,* (**C**) *Mrc1,* (**D**) *Mgl1,* (**E**) *Nos2* mRNA transcript levels. (**F**) Renal active TGFβ protein levels. (**G**) Renal *Fibronectin* and (**H**) *Collagen1* mRNA transcript levels. (**I**) Renal Picro-Sirius-Red (PSR) quantification indicating renal collagen deposition. (**J**) Representative images of PSR staining of renal tissues at day 12 (white bar represent 100 µm). All data are expressed as mean ± SEM, *n* = 6–8 animals per group. * *p* < 0.05, ** *p* < 0.01, *** *p* < 0.001, **** *p* < 0.0001.

## Data Availability

Data are contained within the article and Appendix A.

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
