# Peer review of "NLRX1 Prevents M2 Macrophage Polarization and Excessive Renal Fibrosis in Chronic Obstructive Nephropathy"

_cells, 2023, doi:10.3390/cells13010023_

Round 1
Reviewer 1 Report
Comments and Suggestions for Authors
Ye et al. investigated the role of NLRX1 in macrophage polarization and its association with renal fibrosis in chronic kidney diseases. Their findings indicated that the absence of NLRX1 in BMDMs encourages macrophage polarization, leading to a metabolic reprogramming that favors enhanced OXPHOS-mediated M2 macrophage differentiation. Furthermore, NLRX1 KO mice exhibited pronounced renal fibrosis and injury, with a higher presence of M2 macrophages. While the results are intriguing, they suggest a potential link between NLRX1-mediated macrophage polarization and renal fibrosis. The authors should consider the following critical points:
1. BMDMs may not be the ideal model to examine the impact of NLRX2 on macrophage polarization. It would be more appropriate to isolate macrophages from kidney disease models in both WT and NLRX1 KO mice.
2. To determine the influence of macrophages on renal fibrosis in chronic kidney diseases, techniques like macrophage depletion using liposomes or the application of M2 polarization inhibitors should be employed.
3. When presenting the BMDM macrophage polarization assay, the duration for which the cells were treated with LPS/IFNG or IL4 should be specified.
4. Additional M2 markers, such as CD206 and IL10, should be incorporated to better characterize the M2 macrophages in both in vitro and in vivo settings.
Reviewer 2 Report
Comments and Suggestions for Authors
Kidney dysfunction in chronic kidney disease is mainly caused by renal fibrosis, regardless of the primary renal ailment. Macrophages significantly influence renal fibrosis progression by modulating their metabolic state. The mitochondria-located NLRX1, an innate immune receptor, regulates immune response and cellular metabolism. In the current study, Liu et al. found that NLRX1 deficiency in mouse bone marrow-derived macrophages leads to a shift towards the pro-fibrotic M2 subtype, causing increased renal fibrosis and injury in a UUO model. Therefore, targeting NLRX1 may offer a therapeutic strategy to counteract renal fibrosis in chronic kidney diseases.
The major comments are listed below,
1. Please check all the spells in the text, such as oligomycine, contral, which might lead to confuse of the reader.
2. Please make sure the subtitle of Figures such as Figure B', E' is allowed in this journal.
3. In figure 1, BMDM needs to be induced from bone marrow-derived cells. How can the authors distinguish the differences between WT and KO mice were not due to development? The authors need to provide M0, M1 and M2 markers to prove a successful BMDM induction in both mice.
4. BMDM cannot representative the status of renal macrophages, all the experiments or conclusion should be proved in renal macrophages rather than BMDM.
5. M1, but not M2, leads to renal injury. M2 can lead to fibrosis, but the cause of ischemia injury was mainly due to M1. Herein, the author found the NLRX1 defciency had little effects to M1 polarization but aggravates renal injury, which againsts the findings.
6. All the fibrosis should be stained by Masson trichrome staining and quantified.
7. Does NLRX1 deficiency only has effects to macrophages? The author should confirm the NLRX1 effects to M2 polarization play a role of paramount importance in UUO-induced fibrosis. Cell transfer and depletion should be performed.
Author Response
Kidney dysfunction in chronic kidney disease is mainly caused by renal fibrosis, regardless of the primary renal ailment. Macrophages significantly influence renal fibrosis progression by modulating their metabolic state. The mitochondria-located NLRX1, an innate immune receptor, regulates immune response and cellular metabolism. In the current study, Liu et al. found that NLRX1 deficiency in mouse bone marrow-derived macrophages leads to a shift towards the pro-fibrotic M2 subtype, causing increased renal fibrosis and injury in a UUO model. Therefore, targeting NLRX1 may offer a therapeutic strategy to counteract renal fibrosis in chronic kidney diseases.
The major comments are listed below,
- Please check all the spells in the text, such as oligomycine, contral, which might lead to confuse of the reader.
Answer to Q1: Thanks for your suggestions. The spells have been corrected in the text and figures.
- Please make sure the subtitle of Figures such as Figure B', E' is allowed in this journal.
Answer to Q2: Thanks for the suggestion. All the subtitles of figures have been revised.
- In figure 1, BMDM needs to be induced from bone marrow-derived cells. How can the authors distinguish the differences between WT and KO mice were not due to development? The authors need to provide M0, M1 and M2 markers to prove a successful BMDM induction in both mice.
Answer to Q3: We thank the reviewer for the comments but we have already provided such information in the original manuscript. In Figure 1B-C-D-E indeed we have provided M1 and M2 markers after polarization to prove a successful polarization in BMDM isolated from both mice. We measured the gene expression levels of M1 marker (Nos2) and M2 markers (Arg-1, Mgl-1 and Mrc-1) in both WT and NLRX1 KO BMDMs, including naïve BMDMs (M0). The results showed successful upregulation of relevant macrophage polarization markers in M1 and M2 within WT BMDMs, indicating a successful BMDM induction in mice. In addition, we found that the M2 markers are significantly increased in NLRX1 KO naïve macrophages (M0) when compared to WT cells, suggesting that loss of NLRX1 can steer BMDMs towards a pro-fibrotic M2 phenotype.
- BMDM cannot representative the status of renal macrophages, all the experiments or conclusion should be proved in renal macrophages rather than BMDM.
Answer to Q4: We do agree with the reviewer that BMDMs cannot represent renal macrophages and admit a methodologic limitation in our study due to the use of BMDMs. However, it is technically extremely difficult to isolate macrophages from fibrotic tissue since the extraction process itself can result in a bias towards the preferential extraction of less activated macrophages. Therefore, we used BMDMs in our study to explore the role of NLRX1 in macrophage polarization. Moreover, BMDMs have also been well described and applied in multiple studies about renal injury, inflammation and fibrosis (PMID: 37463911; 34925317; 34337860). We have included this limitation in our discussion in the revised version (Refer to line 386-394).
- M1, but not M2, leads to renal injury. M2 can lead to fibrosis, but the cause of ischemia injury was mainly due to M1. Herein, the author found the NLRX1 defciency had little effects to M1 polarization but aggravates renal injury, which againsts the findings.
Answer to Q5: We completely agree with the reviewer that indeed M1 polarization plays an important role in acute kidney injury induced by renal ischemia reperfusion. In the current manuscript however, we used a different model which is the UUO-induced renal fibrosis, a chronic model of kidney injury, where M2 macrophages are major player, rather than M1. We observed that 3 days after UUO, markers of M1 and M2 macrophages were equally increased in WT mice, compared to CTR (Figure 4C-E manuscript) so we don’t think that the degree of renal injury in this chronic model is dependent on M1 macrophages, as the reviewer suggested. The degree of tubular damage observed in NLRX1 KO mice, as already mentioned, could be a response to TGFβ released from NLRX1 KO macrophages, which show increased M2 phenotype as shown in Figure 4B-C and in our in vitro study. Additionally, many studies have shown that in renal fibrosis, KIM1+ TECs are often surrounded by macrophages and correlates with αSMA+ myofibroblasts (PMID: 33951465). Therefore, we think that our in vivo data do not show any contradictory findings.
- All the fibrosis should be stained by Masson trichrome staining and quantified.
Answer to Q6: Pico Sirius Red (PSR) staining is considered a gold standard and commonly used method to evaluate the development of renal fibrosis. Many investigators, including our group have used PSR staining to quantify renal fibrosis in preclinical models of CKD (PMID: 31354698, PMID: 32518142). We would like to thank the reviewer for the suggestion but we don’t think it is necessary to perform a different staining to show collagen deposition.
- Does NLRX1 deficiency only has effects to macrophages? The author should confirm the NLRX1 effects to M2 polarization play a role of paramount importance in UUO-induced fibrosis. Cell transfer and depletion should be performed.
Answer to Q7: Thanks for your suggestion. It is definitely an interesting idea to perform cell transfer and depletion experiment, but it is beyond the scope of this study. We acknowledge that our study has some limitations and we included this in the Discussion part. (Refer to line 390-394)
Reviewer 3 Report
Comments and Suggestions for Authors
Ye et al. report the protective effect of NLRX1 in UUO-induced renal fibrosis through the regulation of M2 macrophage polarization. The authors have addressed the importance of the role of macrophages’ NLRX1, which regulates the metabolic and functional polarization of macrophages. However, in the UUO model, the authors use constitutional NLRX1 knockout mice instead of macrophage-specific NLRX1 knockout. This experimental work is well-designed and conducted. However, I have two comments on this work that must be addressed adequately.
1. Authors need to address the protective role of macrophage-specific NLRX1 in UUO-induced renal fibrosis by regulation of M2 polarization in vivo conditions. So, the authors need to use the macrophage-specific NLRX1 knockout mice model by using lysozyme Cre and have UUO surgery for their effect on fibrosis. To address NLRX1 as immunometabolic regulator of macrophage polarization, authors need to macrophage specific study.
2. UUO-induced renal fibrosis is usually a slow-progress model that needs 7 to 14 days after surgery. Why do the authors use the 3-day after UUO surgery to address fibrotic changes in the kidney? In Figure 4, the authors use only PSR staining for 12 days after UUO.
Author Response
Ye et al. report the protective effect of NLRX1 in UUO-induced renal fibrosis through the regulation of M2 macrophage polarization. The authors have addressed the importance of the role of macrophages’ NLRX1, which regulates the metabolic and functional polarization of macrophages. However, in the UUO model, the authors use constitutional NLRX1 knockout mice instead of macrophage-specific NLRX1 knockout. This experimental work is well-designed and conducted. However, I have two comments on this work that must be addressed adequately.
- Authors need to address the protective role of macrophage-specific NLRX1 in UUO-induced renal fibrosis by regulation of M2 polarization in vivo conditions. So, the authors need to use the macrophage-specific NLRX1 knockout mice model by using lysozyme Cre and have UUO surgery for their effect on fibrosis. To address NLRX1 as immunometabolic regulator of macrophage polarization, authors need to macrophage specific study.
Answer to Q1: we would like to thanks the reviewer for these relevant suggestions. We agree that further investigations with macrophage-specific NLRX1 knockout mice could help elucidate the macrophage-specific role of NLRX1 in renal fibrosis. We addressed this partly with in vitro study but further research is needed and we include this limitation in the Discussion part. (Refer to line 386-394)
- UUO-induced renal fibrosis is usually a slow-progress model that needs 7 to 14 days after surgery. Why do the authors use the 3-day after UUO surgery to address fibrotic changes in the kidney? In Figure 4, the authors use only PSR staining for 12 days after UUO.
Answer to Q2: We do agree that UUO induced renal fibrosis is a slow process and that 7 and 14 days after surgery, animals will present diffuse fibrosis. On the other hand, alterations induced by NLRX1 absence can also be overshadowed by the complex kidney microenvironment at these time points. Besides, we also found that NLRX1 absence could cause a difference at day3 post-surgery (Figure 4, manuscript), indicating a functional role of NLRX1 in the early stage of UUO and a significant influence in the development of renal fibrosis at a later time point (day 12) as presented by PSR staining.
Reviewer 4 Report
Comments and Suggestions for Authors
This is a well-designed study with sound methodology. It provides an important issue in understanding CKD, development of renal failure and helps identify a gap in the literature. Little change is needed for publication and really only involves the abstract. It best to mention in your abstract that this study is an animal model study. The abstract should also be more structured with appropriate headings that subsequently report on key elements of the study.
Author Response
Thanks for your comments and suggestions. The abstract has been restructured with headings.
Round 2
Reviewer 1 Report
Comments and Suggestions for Authors
The authors have addressed my concerns.
Reviewer 2 Report
Comments and Suggestions for Authors
No more concerns.